Isolation and functional analysis of acid-producing bacteria from bovine rumen

Yu Jinming 1
Li Cunyuan 1
Li Xiaoyue 1
Liu Kaiping 1
Liu Zhuang 1
Ni Wei 1 2
Zhou Ping 2 zhpxqf@163.com
Wang Limin 2
Hu Shengwei 1 2 hushengwei@163.com
1 College of Life Science, Shihezi University , Shihezi, Xinjiang , China
2 State Key Laboratory of Sheep Genetic Improvement and Healthy Production, Xinjiang Academy of Agricultural and Reclamation Science , Shihezi, Xinjiang , China
Sotelo-Mundo Rogerio
Electronic publication date: 2023 Oct 18
Publication date: 2023
Volume: 11
Electronic Location ID: e16294
Received 2023 Jul 10; Accepted 2023 Sep 22
Copyright: © 2023 Yu et al.
Copyright year: 2023
Copyright holder: Yu et al.
License: This is an open access article distributed under the terms of the Creative Commons Attribution License, which permits unrestricted use, distribution, reproduction and adaptation in any medium and for any purpose provided that it is properly attributed. For attribution, the original author(s), title, publication source (PeerJ) and either DOI or URL of the article must be cited.
License URL: https://creativecommons.org/licenses/by/4.0/

Keywords: Bovine rumen, Sheep, Microbes, SCFAs, 16S rRNA

Funding: Sheep Genetic Improvement and Healthy Production 2021ZD08 Third Xinjiang Scientific Expedition Program 2021xjkk0605 and 2021xjkk0504 National Natural Science Foundation of China (NSFC) 32360016 This work was supported financially by the Foundation of State Key Laboratories for Sheep Genetic Improvement and Healthy Production (2021ZD08), the Third Xinjiang Scientific Expedition Program (2021xjkk0605 and 2021xjkk0504) and the National Natural Science Foundation of China (NSFC) (32360016). The funders had no role in study design, data collection and analysis, decision to publish, or preparation of the manuscript.

==============================
Ruminants such as cattle rely mainly on microbes in the rumen to digest cellulose and hemicellulose from forage, and the digestion products are mainly absorbed and utilized by the host in the form of short chain fatty acids (SCFAs). This study aimed to isolate acid-producing strains from the cattle rumen and investigate their functions. A total of 980 strains of acid-producing bacteria were isolated from cattle rumen contents using a medium supplemented with bromocresol green. Combined with the test of acid production ability and 16S rRNA amplicon sequencing technology, five strains were selected based on their ability to produce relatively high levels of acid, including Bacillus pumillus, Enterococcus hirae, Enterococcus faecium, and Bacillus subtilis. Sheep were treated by gavage with a mixed bacterial suspension. The results showed that mixed bacteria significantly increased the body weight gain and feed conversion rate of sheep. To investigate the function of acid-producing bacteria in sheep, we used 16S rDNA sequencing technology to analyze the rumen microbes of sheep. We found that mixed bacteria changed the composition and abundance of sheep rumen bacteria. Among them, the abundance of Bacteroidota, Actinobacteriota, Acidobacteriota, and Proteobacteria was significantly increased, and the abundance of Firmicutes was significantly decreased, indicating that the changes in gut microbiota changed the function of the sheep rumen. The acid-producing bacteria isolated in this study can effectively promote the growth of ruminants, such as cattle and sheep, and can be used as additives to improve breeding efficiency, which lays a foundation for subsequent research on probiotics.

Introduction

As a representative ruminant, the stomach of cattle, a large herbivore, has evolved into a series of four different structures, namely, the rumen, reticulum, omasum, and abomasum. The rumen is the first stomach of ruminants and plays an important role in digestion and absorption. The rumen microbial ecosystem is one of the most complex intestinal microbial ecosystems in animals (Weimer, 2015). The microbial structure of the gastrointestinal tract is complex, and the number of microbes is large, mainly including bacteria, archaea, fungi, and protozoa. Studies have shown that bacteria are the most abundant microbes in the rumen, and adult ruminants have approximately 1011 bacteria per gram of rumen content, including approximately 200 different bacterial species (Newbold & Ramos-Morales, 2020; Russell & Rychlik, 2001; Makkar & Mcsweeney, 2005). Gut microbes play an important role in maintaining the homeostasis of the host digestive tract, nutrient digestion and absorption, production performance, immune health, and behavior (Clemente et al., 2012). The rumen is a natural fermenter of ruminants and a unique organ of ruminants, the biggest feature of which is that the absorption of nutrients is completed through the catabolic effect of microbes in the rumen. The different components of the cattle diet are mainly digested by the rumen. The rumen contains many microbes that can degrade cellulose, hemicellulose, and non-protein nitrogen in the diet to produce metabolites that can be directly used by the host, such as short chain fatty acids (SCFAs). Some of these metabolites are absorbed by rumen epithelial cells, and some are transported to the whole body through blood circulation. Ruminal fermentation can provide 65–75% of the energy for the host, thus ensuring the basic metabolic activities, growth, and development of the host (Duskaev et al., 2021; Anderson et al., 2016; Weston & Hogan, 1968; Jewell et al., 2015; Shabat et al., 2016).

Ruminants mainly feed on dietary fiber, which can produce large amounts of acidic substances such as SCFAs after fermentation. Acids have multiple functions in animals and are involved in several ruminant metabolic pathways. SCFAs, which mainly exist in animals in the form of salt ions, can effectively improve intestinal function, relieve inflammation in vivo, regulate cell proliferation and differentiation, and provide preferential energy sources for colon cells (Spanogiannopoulos et al., 2016; Duncan et al., 2009). SCFAs are important metabolites of gut microbes that provide nutrition and energy to intestinal epithelial cells and improve intestinal function. Acetate and propionate mainly enter the liver through the blood and participate in the metabolic activities of the body. Acetate can be used to synthesize cholesterol and other substances and can also be used directly by peripheral tissues. Propionate is the main substrate for gluconeogenesis and can inhibit fat synthesis. Butyrate, which can be used by epithelial cells, is the most important energy source in the colon and cecum. In addition, studies have shown that SCFAs can promote the colonization of intestinal flora in infants, promote the growth and maturation of intestinal tissues, and play a crucial role in remodeling the composition and function of gut microbes (Liu et al., 2016; Fan & Pedersen, 2021; Bilotta et al., 2020; Sun et al., 2018).

Probiotics as feed additives have multiple functions in humans and animals. First, probiotics can enhance the ability of animals to digest food and promote intestinal health. Second, probiotics can improve immunity in animals, inhibit the growth of harmful bacteria, regulate the balance of intestinal microorganisms, and promote the growth and development of the host. Nowak et al. (2017) found that adding probiotics to pig diets could significantly increase the daily gain and content of SCFAs in pigs. To date, many bacteria have been used as feed additives that can effectively promote the growth and development of animals, such as Bacillus pumillus (Bilal et al., 2021) and Bacillus subtilis (Lefevre et al., 2017).

The global live cattle breeding industry has been growing for a long time. With changes in people’s lifestyles, the demand for beef and milk continues to increase. Improving the growth yield of cattle and reducing production costs have become topics of interest in current research. In this study, we explored ways to improve the growth rate and feed conversion of cattle rumen microbes. In this study, 980 bacterial strains with acid-producing abilities were obtained from cattle rumen contents using a bromocresol green identification medium. Combined with determining acid-producing ability and 16S rRNA amplicon sequencing technology, five strains with strong acid-producing abilities were selected. It was verified that acid-producing bacteria could promote sheep growth, and the cause of weight change in cattle was explored.

Materials and Methods

Sample collection

Cattle rumen contents were collected from Xinjiang Western Animal Husbandry Co., Ltd (Xinjiang, China) and were removed from cattle near the reticulum immediately after slaughter in the abattoir, placed in a sterile 50-mL centrifuge tube, and rapidly stored at 4 °C. After returning to the laboratory, a small portion was removed for the isolation and culture of acid-producing bacteria, and the remainder was stored in a refrigerator at –80 °C.

Isolation and purification of acid-producing bacteria

Briefly, 5 g of rumen contents was accurately weighed into a triangular flask, and 45 mL of phosphate buffer saline (PBS) was added to obtain 10−1 diluent, after evenly shaking, and diluted 10 times with PBS successively to 10−8, with a total of eight gradients (10−1–10−8). After fully and evenly shaking, 100 μL was absorbed for each gradient and evenly coated on bromocresol green identification medium (beef extract 5 g, peptone 5 g, glucose 20 g, NaCl 5 g, bromocresol green 0.1 g, agar 20 g, pH 7.0, 1 L water), and aerobic culture was performed. Two replicates were performed for each gradient. After 18 h of culture, colonies that produced yellow transparent rings were selected and cultured in a tryptic soy broth (TSB) liquid for shaking culture. After 12 h, streaking purification was performed using a TSB solid medium. After three rounds of purification, a pure culture was obtained.

Molecular identification of acid-producing bacteria

The 16S sequences of the strains were amplified using universal 16S primers 27F (5′-GAGTTTGATCTGGCTCAG-3′) and 1492R (5′-ACGGCTACCTTGTTACGACTT-3′) (Koeck et al., 2014). The PCR reaction system comprised 10 μL of PrimeSTAR MaxPremix (2×), 0.5 μL of forward and reverse primers, 0.5 μL of bacterial solution, and 8.8 μL of ddH2O. The PCR reaction conditions were as follows: 95 °C for 5 min, 35 cycles of 95 °C for 15 s, 56 °C for 20 s, 72 °C for 30 s, and 72 °C for 3 min. After the PCR reaction, 1% agarose gel electrophoresis was used to detect the PCR products. When a single band appeared, and its size was the same, the PCR products were sent to Sangon Bioengineering (Shanghai, China) Co., LTD for sequencing analysis. The sequencing results were compared with Blastn in the BLAST platform in NCBI, and a phylogenetic tree was constructed using MEGA software.

Identification of acid production capacity

Five strains of acid-producing bacteria were seeded in bromocresol green screening medium, and bromocresol green changed from green to yellow when exposed to acid. Acid-producing strains were identified based on the size of the yellow area and depth of color in the solid medium.

Fermentation culture of acid-producing bacteria and preparation of mixed bacteria

To obtain a sufficient concentration of bacteria to complete the animal experiments, the obtained acid-producing strains were cultured by shaking flask fermentation using a TSB liquid medium. The seed liquid was inoculated in a TSB fermentation medium according to the inoculum amount of 10%, fermented, and cultured at 37 °C at 180 rpm for 48 h. Finally, the obtained fermentation broth was centrifuged at 4 °C and 5,000 rpm for 5 min to remove the supernatant, and the precipitate contained the required bacteria. The precipitated bacteria were washed twice with PBS buffer to remove the residual medium completely, and the precipitate was suspended in an appropriate amount of PBS to make a high-concentration bacterial suspension; then, 25% pure glycerin was added to it, and it was stored in a refrigerator at –80 °C. The viable count of the preserved bacterial suspension was determined before the gavage treatment of the sheep. Each of the five bacterial suspensions was diluted 10-fold. The bacterial suspensions with three dilutions of 10−8, 10−9, and 10−10 were coated on a TSB solid medium, and each gradient was repeated in three groups. Bacteria were cultured in an incubator at 37 °C for 12 h. The concentration of the high-concentration bacterial suspension was calculated according to the colony forming units, and the concentrations of the five bacterial suspensions were diluted to 1011 cfu/mL. One milliliter of each of the five diluted bacterial suspensions was mixed at equal concentrations to prepare a mixed bacterial suspension.

Animal experiment

Fourteen 6-month-old female sheep were selected as experimental animals and randomly divided into two groups (n = 7). During the experiment, one of the hind legs of each sheep was injured and removed. All the sheep were kept under the same conditions, at Shihezi southwest farm, and the sheep were fed once every morning and evening, and all the sheep were free to eat; the main grain was corn (Table S1). The sheep were weighed before the start of the experiment and every 3 days during the experiment. During the 18-day trial period, the sheep were treated daily by gavage. Each sheep in the experimental group was given 5 mL of mixed bacterial suspension, which contained five kinds of acid-producing bacteria, and the number of viable bacteria was 5 × 1011 cfu (Hamdon et al., 2022), whereas the control group was given the same volume of PBS. The feed intake of the sheep was measured during the experiment, and the feed conversion ratio was calculated (feed conversion rate = sheep daily gain/feed intake). Twenty-four hours after the last gavage treatment, rumen contents near the reticulum end of all sheep were collected using an in vivo rumen fluid collection device and flash-frozen in liquid nitrogen. Frozen samples were sent to NovoGen Co., Ltd. for sequencing. All sheep were raised by local herders, and they were not killed after the experiment. This study was approved by Biology Ethics Committee of Shihezi University (Approval Number: SUACUC-08032).

Analysis of species diversity

We used the CTAB method to extract microbial DNA from bovine rumen. The steps were as follows: 1 mL of CTAB buffer, 20 μL of lysozyme, and 100 mg of rumen contents were mixed evenly in an EP tube and incubated at 65 °C for 2–3 h. After incubation, 950 μL supernatant and 950 μL phenol-chloroform-isoamyl alcohol (25:24:1) solution were collected and mixed evenly in an EP tube and centrifuged at 12,000 rpm for 10 min. The supernatant was collected and evenly mixed with an equal volume of isoamyl chloroform (24:1) in an EP tube. The mixture was then centrifuged at 12,000 rpm for 10 min. The supernatant was collected, and 3/4 volume of isopropyl alcohol was added. The supernatant was stored at –20 °C for 20 min and centrifuged at 12,000 rpm for 10 min. The supernatant was discarded, and the cells were washed twice with 75% ethanol. After drying, the DNA was dissolved in 50 μL sterile water, and RNase A was used to remove RNA (Li et al., 2023). The purity of the DNA samples was detected using agarose gel electrophoresis (1%), and the concentration was diluted to 1 ng/μL using sterile water. Using the extracted DNA as a template, the V4 region (515F:5′-GTGCCAGCMGCCGCGGTAA-3′; 806R:5′-GGACTACHVGGGTWTCTAAT-3′) was amplified by PCR using PrimeSTAR HS DNA Polymerase identical to 16S full-length, and the PCR products were detected by 2% agarose gel electrophoresis. Target bands were recovered using a gel recovery kit (Qiagen, Hilden, Germany).

The NEBNext® Ultra™ II DNA Library Prep Kit was used to construct the library, and Qubit and Q-PCR were used to quantify the constructed library. After the libraries were qualified, NovaSeq6000 was used for sequencing. After cutting off the primer sequence and barcode of the sequencing results, FLASH software (Magoč & Salzberg, 2011) was used to assemble the reads of the samples to obtain raw tags. Subsequently, fastp software was used for quality control of the raw tags to obtain high-quality clean tags with default parameters. Finally, Usearch software was used to compare clean tags with the Silva database to remove chimeras (Haas et al., 2011) to obtain the final effective data, i.e., effective tags. To obtain the final amplicon sequence variants (ASVs), the obtained effective tags were analyzed using the DADA2 module in the QIIME2 software to reduce noise and filter out fragments less than 5. The classification sklearn module in QIIME2 software was used to annotate the species information.

The alpha diversity index was calculated using QIIME2 software, which was used to assess microbial diversity in the samples. UniFrac distances were calculated using QIIME2 software, and PCoA maps were generated using R software (R Core Team, 2013) to compare the diversity of microbes between different samples.

Statistical analysis

Analysis of variance (ANOVA) in this experiment was performed using the statistical package (SPSS 19.0; SPSS, Inc., Armonk, NY, USA), and differences between treatments were considered significant at p < 0.05.

Results

Isolation and identification of acid-producing bacteria

A total of 980 bacterial strains were selected from the yellow region of the solid medium using bromocresol green identification medium. To obtain pure cultures of individual strains, these single colonies were line-purified three times, and pure cultures of these strains were obtained.

To effectively identify the acid-producing strains, we inoculated them in a solid medium supplemented with bromocresol green (Fig. 1). Five strains, numbered 32, 80, 82, 150, and 898, formed yellow circles on the solid medium, indicating their acid-producing ability. We compared the full-length 16S rRNA sequence of the acid-producing bacteria using the BLAST platform and combined it with a phylogenetic tree. We determined that strain 32 was the closest relative to Bacillus pumilus strain LG 135. Strain 80 was the most closely related to the Enterococcus hirae strain ZZU A2. Strain 82 was the most closely related to the Bacillus subtilis strain MK736123.1, strain 150 was the most closely related to the Enterococcus faecium strain CAU10327, and strain 898 was closely related to the Bacillus subtilis strain SLL2 (Fig. 2).

Figure 1 Identification of acid-producing bacteria.

Numbers indicate the acid-producing strains isolated, and yellow areas may indicate the acid-producing capacity of the strains.

Figure 2 Phylogenetic tree of isolated and purified acid-producing strains.

The numbers in the figure indicate the isolated strains, and other strains were obtained from NCBI. Pink for Bacillus subtilis, purple for Bacillus pumillus, green for Enterococcus hirae, blue for Enterococcus faecium, and yellow for other strains.

Mixed bacteria promoted sheep growth

To verify the function of acid-producing bacteria, sheep were selected as experimental animals for functional verification. After gavage treatment, the sheep in the experimental group showed more growth than those in the control group. On the 15th day of gavage treatment, the body weight gain of the experimental groups significantly improved (p = 0.01) (Fig. 3A). Next, we measured the feed conversion rate of the sheep to determine the cause of weight gain. The results showed that the feed conversion rate of the experimental group was 12.01% and that of the control group was 7.75%, indicating that the feed conversion rate of the sheep significantly improved after gavage (p = 0.04) (Fig. 3B). This indicates that the five strains of acid-producing bacteria can improve the growth rate of sheep by increasing the feed conversion rate.

Figure 3 Effect of mixed acid-producing bacteria on growth performance of sheep.

(A) Weight changes in sheep treated with mixed acid-producing bacteria (p < 0.05). The abscissa represents the time of exertion of the mixed bacteria on the sheep, and the ordinate represents the change in the weight of the cotton sample (kg). (B) Changes in feed conversion in sheep treated with mixed bacteria.

Mixed bacteria changed the composition of the microbiota in the sheep rumen

16S rRNA sequencing was performed to explore further potential interactions between the effects of mixed bacteria on body weight gain in sheep and the gut microbiota of the sheep rumen. In total, 946,145 reads, with an average of 78,845 reads per sample, were generated for the raw data. A total of 752,227 reads, with an average of 62,685 reads per sample, were generated for the richness analysis. After annotating the species information, we selected the top 10 abundances at both taxonomic levels, i.e., phylum and class, for the histogram (Figs. 4A and 4B). Firmicutes, Bacteroidetes, Proteobacteria, Patescibacteria, Acidobacteriota, and Actinobacteria were dominant at the phylum level. Compared to the control group, the abundance of Firmicutes, Cyanobacteria, and Patescibacteria decreased. The abundance of Bacteroidetes, Proteobacteria, and Actinobacteria increased. Clostridia, Bacteroidia, Gammaproteobacteria, Saccharimonadia, Bacilli, Actinobacteria, and Acidobacteria were dominant at the class level. Their abundance in the sheep rumen changed. According to the species annotation and abundance information for all samples at the genus level, the top 35 genera with the highest abundance were selected. According to the abundance information for each sample, the genera were clustered at the species and sample levels, and a heat map was drawn to determine the aggregation content of species in each sample (Fig. 4C).

Figure 4 Intestinal microbial classiûcation composition of rumen of sheep in each group.

Relative abundance map of species taxonomic at the levels of phylum and class: (A) phylum level; (B) class level. (C) Heatmap of fecal samples at the level of genus. NC was the control group, TG was the mixed bacteria treatment group.

The boxplot effectively reflects the species differences between groups (Fig. 5A). Kruskal-Wallis analysis showed that the microbial diversity index of the experimental group was significantly higher than that of the control group (p = 0.004), indicating that the microbial richness in the rumen of sheep was significantly improved after gavage with mixed bacteria. Principal coordinate analysis (PCoA) was used to analyze the changes in the structure of the bacterial communities in the rumen samples of the two groups and to draw PCoA diagrams (Fig. 5B). The results showed that the structure of the intestinal flora changed after 18 days of gavage, indicating that there was a large difference in microbial communities between the two groups.

Figure 5 Analysis of bacteria species richness in sheep rumen.

(A) Alpha diversity index between experimental and control groups after mixed bacteria treatment (p < 0.05). (B) PCoA diagram of microbial the rumen of diûerent sheep after treatment with mixed bacteria. Each dot in the ûgure represents a sheep sample, red for the control group and green for the mixed bacterial treatment group. The closer the distance between samples, the closer the microbial species structure in rumen.

Using the t-test, we found species with significant differences (p < 0.05) in different groups at the genus level and drew the volcano map (Fig. 6A). There were 50 significantly different genera between the control and experimental groups, including 35 significantly downregulated and 15 significantly upregulated genera. The LEfSe method (Segata et al., 2011) was used to analyze the species with significant differences between the two groups (Fig. 6B), and it was found that there were significant differences in the dominant microbes between the two groups. Firmicutes played an important role in the control group. Additionally, the abundance of p_Firmicutes decreased, whereas that of p_Bacteroidota, p_Actinobacteriota, p_Acidobacteriota, and p_Proteobecteria increased in the sheep rumen.

Figure 6 Analysis of species diûerences among microbiomes in sheep rumen.

(A) Volcano diagram. Each point in the figure represents a differential species, where up represents the higher abundance of this differential species in the experimental group than in the control group, and the opposite is true for Down. (B) Evolutionary branch map. In the evolutionary clade map, the circles radiating from inside to outside represent taxonomic levels from phylum to genus. Each small circle at a different taxonomic level represents a classification at that level, and the small circle diameter size is proportional to the relative abundance size. Coloring principle: the species with no significant difference is uniformly colored yellow, and the different species are colored according to the group. The blue node represents the microbial group that plays an important role in the red group, and the green node represents the microbial group that plays an important role in the green group. If a group is missing, it means that there is no significant difference in the group, so the group is missing.

Discussion

The rumen is unique to ruminants, and many microbes in the rumen can effectively promote host growth. The food consumed by ruminants, such as cattle and sheep, is rich in large amounts of cellulose and hemicellulose, and the decomposition of cellulose and hemicellulose is mainly completed by microbes in the rumen. Gut microbes break down these hard-to-use substances into substances easily utilized by the host, such as SCFAs, methane, and microbial proteins (Fan et al., 2020).

In this study, according to the principle that bromocresol green turns yellow when it reacts with acid (Adeoye & Lateef, 2021), we isolated strains with high SCFA production from cattle stomach contents under aerobic conditions using a medium supplemented with bromocresol green. In this study, 980 strains of acid-producing bacteria were isolated using bromocresol green screening medium, and their acid-producing abilities were identified. Five strains with strong acid-producing abilities were selected and identified by molecular identification and a phylogenetic tree: Bacillus pumilus, Enterococcus hirae, Enterococcus faecium, and two strains of Bacillus subtilis. These strains are common probiotics usually made into dry powder preparations and used as feed additives in daily production. In this study, we prepared a mixed bacterial suspension of five strains and treated sheep by gavage. The results showed that the mixed bacterial suspension could significantly improve the daily weight gain of sheep, indicating that the five isolated strains could effectively improve the gastrointestinal tract’s digestive ability and nutrient absorption efficiency. The significant change in the feed conversion rate also effectively demonstrates this point. In recent years, probiotics have been widely used in livestock breeding, and mixed bacterial preparations have attracted increasing attention. Zhang et al. (2022) found that adding Bacillus subtilis and Enterococcus faecium to the diet of finishing pigs could effectively reduce the feed-to-meat ratio and improve the growth rate of finishing pigs. Wang et al. (2021) reported that feeding lactating sows feed co-fermented with Bacillus subtilis and Enterococcus faecium increased feed intake and changed meat quality; however, it improved the daily gain of piglets and enhanced their immunity. Wei et al. (2020) reported that a strain of Enterococcus hirae isolated from healthy infants could improve the symptoms of type 2 diabetes mellitus in rats, improve glucose tolerance, and reduce total bile acid content in rats. Zhang, Wang & Wei (2020) showed that adding Bacillus amyloliquefaciens and Bacillus pumilus to the feed of weaned goats could effectively promote the development of the rumen and small intestine of weaned goats, increase the abundance of probiotics, and reduce the abundance of pathogenic bacteria.

Through 16S rRNA sequencing analysis of sheep rumen contents, we found that the abundance of Bacillus pumilus, Enterococcus hirae, Bacillus subtilis, and Enterococcus faecium did not change, but the composition of bacteria in the rumen did change, which is similar to the results of Wang et al. (2022). We performed Chao1 index analysis and PCoA cluster analysis of the microbial communities between the two groups and found that Firmicutes and Bacteroidota were the most abundant phyla in the rumen of healthy sheep, which is consistent with previous studies (Hu et al., 2021; Liu et al., 2021). After 18 days of gavage, the results showed that there were significant changes in the rumen microbial community in the treatment group, and the rumen microbial species richness of the experimental group was significantly higher than that of the control group. Increasing species diversity helps rumen microbes maintain homeostasis and adapt to a changing environment. Moreover, it can also improve nutrient utilization and feed conversion rates.

By using ASV species annotation and t-test methods, we found that the species of microbes that play a major role in sheep rumen changed, with p_Bacteroidota, p_Actinobacteriota, p_Acidobacteriota, and p_Proteobecteria playing the main roles in the treatment group. The abundance of Bacteroidota increased significantly after the gavage treatment, and Bacteroidota became the most abundant phylum in the rumen. Bacteroidota in the gut are often considered beneficial because they degrade starch and polysaccharides in the rumen and are the main producers of SCFAs in the gut. Its products are mainly acetate, propionate, and butyrate (Rosewarne et al., 2014; Ahmad et al., 2020; Belanche et al., 2019), and the energy source of ruminants depends mainly on the absorption of SCFAs. Therefore, mixed bacteria can effectively increase the energy utilization rate of the host. Wang et al. showed that the human body is rich in Bacteroidota and is expected to become the next generation of probiotics (Cui et al., 2022). Acidobacteriota widely exist in nature, among which Acidobacteria are the most abundant in the soil. However, it is difficult to cultivate Acidobacteriota, which have a good growth state in acidic environments (Sikorski et al., 2022). Studies have shown that Acidobacteriota can effectively degrade cellulose, hemicellulose, and xylan in nature (Belova et al., 2022). Therefore, we believe that mixed Acidobacteriota can increase the abundance of Acidobacteriota in the rumen by changing the pH of the sheep rumen, which increases the degradation efficiency of cellulose and hemicellulose in the feed, and effectively improving the feed conversion rate of sheep and promoting sheep growth. The presence of Proteobacteria is closely related to the health of mammals. Studies have shown that the lower respiratory tract of felines is mainly composed of Proteobacteria. When felines suffer from acute or chronic asthma, the abundance of Proteobacteria will decrease sharply (Vientós-Plotts et al., 2022). Actinobacteriota are widely used in the fields of medicine and biotechnology, and Actinobacteria are important sources of bioactive chemicals and major producers of currently used antibiotics, contributing two-thirds of the clinical antibiotics and a variety of industrial enzymes (van Wezel et al., 2006). Therefore, we believe that the gavage of sheep with mixed bacteria can change the living environment of microbes in the sheep rumen, increase the type and abundance of probiotics, better degrade cellulose and hemicellulose in the sheep rumen, produce SCFAs, improve the efficiency of energy absorption by the host, and increase the feed conversion rate. Moreover, an increase in probiotics can improve the body’s immunity, such as an increase in the abundance of actinomycetes, which can cause an increase in the content of antibiotics in the intestine and help to improve the host’s resistance to harmful bacteria.

Conclusions

In this study, 980 strains of acid-producing bacteria were isolated using a bromocresol green screening medium and their acid-producing ability was determined, and five strains with strong acid-producing abilities were selected, including Bacillus pumillus, Enterococcus hirae, Enterococcus faecium, and Bacillus subtilis. The five strains were mixed to obtain a bacterial suspension, with the concentration of each strain being 1,011 cfu/mL, and healthy sheep were treated by gavage. The results showed that the mixed bacteria could increase the daily gain and feed conversion rate of sheep and change the composition and abundance of microorganisms in the sheep rumen. The strains isolated in this study are common probiotics with significant growth-promoting effects, which lay the foundation for further studies on the functions of probiotics. Given the significant effect of mixed bacteria, we can prepare dry powder preparations of mixed bacteria and use them as feed additives to reduce the cost of animal husbandry and improve breeding efficiency.

Supplemental Information

Supplemental Information 1 Author Checklist.

Click here for additional data file.

Supplemental Information 2 Feed formula.

Click here for additional data file.

Supplemental Information 3 Raw data from animal experiments.

Click here for additional data file.

Additional Information and Declarations

Competing Interests

Author Contributions

Animal Ethics

Data Availability

The authors have declared that no competing interests exist.

Jinming Yu conceived and designed the experiments, performed the experiments, analyzed the data, authored or reviewed drafts of the article, and approved the final draft.

Cunyuan Li conceived and designed the experiments, performed the experiments, prepared figures and/or tables, authored or reviewed drafts of the article, and approved the final draft.

Xiaoyue Li conceived and designed the experiments, performed the experiments, analyzed the data, prepared figures and/or tables, authored or reviewed drafts of the article, and approved the final draft.

Kaiping Liu conceived and designed the experiments, performed the experiments, analyzed the data, authored or reviewed drafts of the article, and approved the final draft.

Zhuang Liu conceived and designed the experiments, performed the experiments, authored or reviewed drafts of the article, and approved the final draft.

Wei Ni conceived and designed the experiments, analyzed the data, authored or reviewed drafts of the article, and approved the final draft.

Ping Zhou conceived and designed the experiments, analyzed the data, authored or reviewed drafts of the article, and approved the final draft.

Limin Wang conceived and designed the experiments, analyzed the data, authored or reviewed drafts of the article, and approved the final draft.

Shengwei Hu conceived and designed the experiments, analyzed the data, prepared figures and/or tables, authored or reviewed drafts of the article, and approved the final draft.

The following information was supplied relating to ethical approvals (i.e., approving body and any reference numbers):

Biology Ethics Committee of Shihezi University

The following information was supplied regarding data availability:

The datasets generated during the current study are available from NCBI Sequence Read Archive (SRA): PRJNA983066.

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
