# Peer review of "Isolation and functional analysis of acid-producing bacteria from bovine rumen"

_PeerJ, doi:10.7717/peerj.16294_

## Round 0.1 · original submission · Minor Revisions

Please submit a thoroughly revised version of your rebuttal letter, taking into account the editorial comments.

**Language Note:** PeerJ staff have identified that the English language needs to be improved. When you prepare your next revision, please either (i) have a colleague who is proficient in English and familiar with the subject matter review your manuscript, or (ii) contact a professional editing service to review your manuscript. PeerJ can provide language editing services - you can contact us at copyediting@peerj.com for pricing (be sure to provide your manuscript number and title). – PeerJ Staff

Reviewer 1 ·

Basic reporting

Yu et al.'s manuscript isolates acid-producing bacteria from bovine rumen and test their probiotic potential with animal experiments and further 16S rRNA analysis.
The manuscript has a well-defined question with an adequate experimental and analysis design. Therefore, I only have a few minor changes that should be addressed before accepting the manuscript for publication.

Experimental design

The experimental design and statistical analyses are adequate to answer the main question. A few details are missing, and I address them in the Additional comments.

Validity of the findings

The results interpretation is adequate and in line with the main question. All the data provided supports the results and conclusions

Additional comments

1. To my understanding, “bovine” refers to animals such as cows, buffalos and bison. And ovine refers to sheep. Was the analyzed rumen from cows while the animal experiments were performed on sheep? Please clarify and change the terms accordingly.
2. Specify the identity and coverage results of the Blast of the complete 16S rRNA gene analysis to identify the acid-producing bacteria.
3. Were all the sheep from the same sex?
4. Please clarify in the methods section the total duration of the gavage treatment. This detail is only suggested once in the Results and Discussion section.
5. Please include the references or the rational used to decide de bacterial dosage used in the gavage treatment.
6. The CTAB method for DNA extraction seems appropriate for complex samples such as rumen. However, are there reports that use this method for microbiota analysis. Please include a few lines about this.
7. Please clarify if you used the same enzyme to amplify the complete 16S rRNA gene and the V4 region.
8. Specify the database used for the taxonomic assignation in the V4 region analysis.
9. Clarify the sequencing depth obtained for all samples in the V4 analysis. And specify what the sequencing depth used for the richness analysis was.
10. Specify the quality parameters used to preprocess the V4 sequencing reads.
11. Specify the formulas used to calculate the feed conversion rate.
12. In line 320 is mentioned that “5 strains with strong acid-producing ability…”. What parameters are the authors using to characterize these strains as “strong”?
13. Line 360 change the word AVS to ASV

·

Basic reporting

All the comments are included in a separate file

Experimental design

All the comments are included in a separate file

Validity of the findings

All the comments are included in a separate file

---

## Round 0.2 · accepted · Accept

Thank you for addressing all of the requested revisions and corrections. Your manuscript has now been accepted by PeerJ.